# Mechanisms of Producing Primordial Black Holes and Their Evolution

**Maxim A. Krasnov * and Valery V. Nikulin**

Department №40 "Elementary Particle Physics", Moscow Engineering Physics Institute, National Research Nuclear University MEPhI, Kashirskoye Shosse 31, 115409 Moscow, Russia
* Correspondence: morrowindman1@mail.ru

**Abstract:** Primordial black holes have become a highly intriguing and captivating field of study in cosmology due to their potential theoretical and observational significance. This review delves into a variety of mechanisms that could give rise to PBHs and explores various methods for examining their evolution through mass accretion.

**Keywords:** accretion; phase transition; $f(R)$-gravity; primordial black holes; cosmology; domain wall

## 1. Introduction

The idea of primordial black hole (PBH) formation in the early universe was suggested in 1967 by Zeldovich and Novikov [1] and also by Hawking and Carr [2–4]. PBHs have been a subject of interest for fifty years. In particular, only PBHs, by construction, can be light enough for the so-called Hawking radiation to be essential to change the mass of a black hole. Today, the primordial origin of some discovered black holes (quasars at small $z$ [5,6], BHs of intermediate masses detected by gravitational-wave observatories [7]) is hotly discussed [8,9]. It is also worth noting that black holes of primordial origin can be dark matter candidates [10]. At the moment, constraints are imposed on a wide range of masses of a PBH as a dark matter candidate [11]. It has also been shown [12–14] that primordial black hole formation mechanisms are able to form clusters of PBHs and then, some constraints imposed on black holes as dark matter candidates should be reconsidered [13,15,16].

Many different mechanisms for the formation of PBHs have been proposed: density fluctuations [2,3], first-order phase transitions [4,17], cosmic string collapse [18], the appearance of PBHs in hybrid inflation models [19], second-order phase transitions [20,21], and reviews of PBH formation mechanisms are given in [22,23]. Recently, a new mechanism of formation of PBHs discovered in the multidimensional modified gravity model [24,25], containing tensor and quadratic-to-scalar curvature corrections, has been considered. In addition, a modified gravity offers solutions to many cosmological problems [26–29], thus the possibilities of $f(R)$-gravity have been widely studied [30,31].

The next aspect to be discussed is the accretion process. Accretion is the process in which a black hole can capture particles from a nearby source of fluid, and there is an increase in mass as well as angular momentum of the accreting object, see, e.g., [32]. There are different approaches to study this process and some of them are reviewed here. The most common assumption to calculate accretion rate is a spherical symmetry. This assumption is used in papers such as [33–36]. There have also been numerical estimations with relativistic hydrodynamics [37] and their results indicate that mass growth through radial accretion might be significant. It is also necessary to mention accretion disk models. The first realistic model of accretion disks around black holes was formulated in 1973 by Shakura and Sunyaev [38]. This approach was based on the consideration of matter rotating in circular Keplerian orbits around the compact object, losing angular momentum because of the friction between adjacent layers and spiralling inwards. In this process, gravitational energy was released, the kinetic energy of the plasma increased and the disk heated up,

emitting thermal energy into space. For a relativistic analysis, see, e.g., [39]. Various models of accretion disks are considered in [40].

The work plan is organised as follows: In Section 2, we consider the mechanisms of black hole generation, including the traditional Zel'dovich and Novikov mechanism and various types of phase transitions. In Section 3, we discuss the mechanisms of mass accretion. We start with the conventional Bondi solution and one of its applications. Next, we consider a common marginal estimation of mass accretion, i.e., the Eddington limit. The section ends with a discussion of approaches based on the general theory of relativity. In Section 4, we give a brief description of the models considered in this review.

## 2. Production Mechanisms

### 2.1. Initial Density Inhomogeneities

In this section, a brief review of [3] is presented.

This mechanism is based on the collapse of primordial inhomogeneities in a hot plasma and arises within the framework of the standard Big Bang cosmology. Consider a region of the universe of radius $R$. It has a potential energy of self-gravitation

$$\Omega \sim -\rho^2 R^5 \tag{1}$$

ans also a kinetic energy of expansion

$$T \sim \rho R^3 \dot{R}^2, \tag{2}$$

where $\rho$ is the energy density. At the radiation-dominated universe, the pressure and energy density are proportional to $R^{-4}$ since $p = \frac{1}{3}\rho$.

If the density is high enough in a region, gravitational forces may overcome the kinetic energy of expansion and pressure forces. As a result, the cosmological expansion is halted in that region. To overcome pressure forces, the gravitational energy must be greater than the internal energy, which, for $p = \frac{1}{3}\rho$, is $U \sim \rho R^3$, so the necessary condition for collapse is

$$\rho R^2 > \sim 1. \tag{3}$$

When $p \sim \rho_0 \ln(\rho/\rho_0)$, which corresponds to a logarithmic-corrected power-law fluid, $U \sim \rho_0 R^3 \ln(\rho/\rho_0)$, where $\rho_0 \approx 10^{14} \, \text{g/cm}^3$. Thus, necessary condition for the collapse is

$$\rho R^2 > \sim \frac{\rho_0}{\rho} \ln(\rho/\rho_0). \tag{4}$$

This limit is just Jean's length for a radiation-dominated universe.

A drawback of this model is that the mass spectrum in this model is close to monochromatic, so it is impossible to explain the existence of black holes of various masses. It is also impossible to produce clusters of PBHs within this model. This model does not contain a mechanism for the production of compact clusters, and consequently, it is not applicable to explain the merger rates observed by LIGO/VIRGO [9]. It must be stated that this is the first and "conventional" mechanism for black hole production. It does not require any assumption beyond standard Big Bang physics.

### 2.2. First-Order Phase Transitions

Let us consider the mechanism for producing PBHs by means of scalar field dynamics. First-order phase transition as a mechanism to produce black holes was proposed by Khlopov et al. [41]. For a realization of this mechanism, it is necessary that the field potential has at least two minima, and one of them must be false. Let the field be in a false vacuum $\phi_1$ at the initial moment of time, and let us denote the true vacuum by $\phi_0$ (Figure 1). As a result of quantum tunnelling, in one region of space, the field will have

a value of $\phi_1$ and in another region of space, $\phi_0$. These regions are called bubbles. In this formulation, the free energy of a bubble consists of two parts, the volume and the surface energy. We denote the surface energy density $\mu$ and the difference of potential values at minima $\Delta V = E(\phi_0) - E(\phi_1)$. In this case, the free energy of a bubble with radius $R$ and the surface energy density $\mu$ can be written as

$$F(R) = 4\pi R^2 \mu - \frac{4\pi}{3} R^3 \Delta V. \tag{5}$$

Obviously, the dependence (5) has a maximum at the point $R_{cr} = 2\mu/\Delta V$, after which it becomes energetically advantageous for the bubble to expand infinitely. Then, the expansion of bubbles of true vacuum in the region of false vacuum becomes possible, while the potential energy of the false vacuum is converted into a kinetic energy of the walls, which leads to an ultrarelativistic speed of expansion in short time.

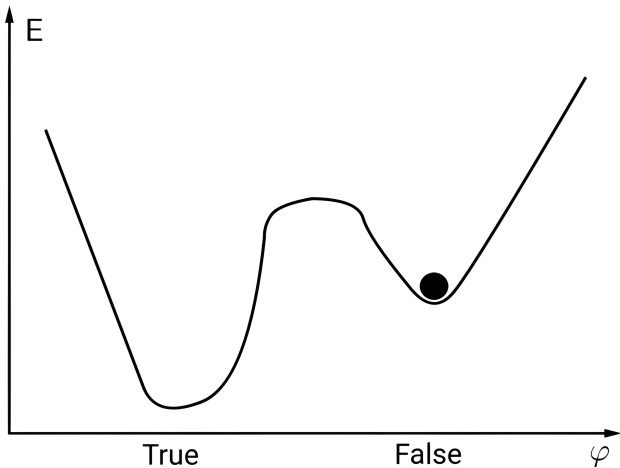

**Figure 1.** Schematic representation of a scalar field potential in which phase transitions of the first kind are possible.

When a pair of bubbles of true vacuum collide, a new bubble of false vacuum can arise in the region of false vacuum. If its size is smaller than its gravitational radius $r_g = 2GM$ ($G$ is a gravitational constant), it becomes a black hole for a distant observer. However, if the bubble shrinks to a size comparable to the wall thickness $d \sim \delta > r_g$, the collapse in the PBH will not occur, since the bubble will oscillate, lose energy, and finally decay by tunnelling.

For some detailed analysis how PBHs could be formed within this mechanism see, e.g., [42,43] and the references therein.

The advantage of this mechanism is a broad mass spectrum [44]. However, one must pay attention to its drawbacks, e.g., it leads to strong inhomogeneities [45], and there is no possibility to form clusters of PBHs. This model also does not contain a mechanism for compact cluster production.

### 2.3. Second-Order Phase Transitions

In contrast to first-order phase transitions, in this mechanism, we do not face collisions of growing domain walls. The closed domain walls shrink and eventually collapse to form black holes. For a realization of this mechanism, it is necessary that the field potential possesses at least two vacuums of the same energy.

The idea is to generate domain walls which could collapse into a PBH after crossing the cosmological horizon. There are two ways to implement this. The first case is based on spontaneous symmetry breaking. The second case is based on the inflationary stage quantum fluctuations.

Let us qualitatively review the first case. Its key feature is a spontaneous symmetry breaking with a consequent change of potential. After the temperature drops below a certain level, the potential acquires several equal-energy minima. This can lead to the generation of shrinking domain walls; see, e.g., [20], where the case of spontaneous symmetry breaking is considered.

It is necessary to mention a model elaborated by A. Dolgov in 1993 [46] with a log-normal mass spectrum. The proposed model was the first where the cosmological inflation and Affleck–Dine baryogenesis [47] were applied to the PBH formation. Mass distribution of PBHs was given by the expression:

$$\frac{dN}{dM} = \mu^2 \exp\left[-\gamma ln^2(M/M_0)\right], \tag{6}$$

where $\gamma$ is a dimensionless constant, according to [46]. It was predicted that $M_0 \sim 10 M_\odot$ [48] and this result was in good agreement with observations [11,49]. This is the only mass spectrum of PBHs which was tested by observations and found in good agreement with the model.

The second mechanism is based on multiple quantum fluctuations generated during the inflationary stage. There is no symmetry breaking in this case, but such mechanism also generates closed domain walls.

Let the field be near the maximum of the potential at the initial moment of time. Classical motion of the field would lead to the field rolling into one of the minima; however, as a result of multiple quantum fluctuations at the inflationary stage (in a field with frozen classical motion), the field can gradually flip over the maximum. As a result, it rolls into different minima in different regions of space. These distinct regions are connected by closed domain walls. The shrinking of such domain walls can lead to the formation of PBHs. One of the first studies dedicated to the elaboration of this mechanism of PBH was performed in [20].

In this case, there is a very broad mass spectrum of PBHs, see, e.g., [13,50]. Nonetheless, the mass spectrum in that case strongly depends on the initial field value, so fine-tuning is required, although fine-tuning is a common problem for inflation theories [51].

There are common features in these two approaches. The characteristic scale of nonvanishing fluctuation at the inflationary stage is $H_{\text{inf}}^{-1}$. If it is formed at the moment $t'$ during inflation so by the end of the inflation-dominated phase, it would be $e^{N_{\text{inf}} - H_{\text{inf}} t'}$ times bigger. Its further evolution depends on the relation between two timescales: $t_\sigma = 1/2\pi G\sigma$, where $\sigma$ is the surface energy density of the wall and $G$ is, again, a gravitational constant, and the moment at which the wall crosses the cosmological horizon, $t_H$.

If $t_\sigma \gg t_H$, the wall is called subcritical and a black hole forms much earlier, before the wall can become dominant in the universe. However, in the case where $t_\sigma \lesssim t_H$, the wall is called supercritical, and there is a wormhole formation as a way to "export" the problem of wall domination to a baby universe; see [13,52–54] and references therein for a detailed analysis.

The advantage of this mechanism is the same as mentioned above, a broad mass spectrum, and it could also be coupled with external processes, e.g., baryogenesis [46]. In contrast with the case considered above, there is no explicit mechanism to form compact clusters of black holes.

### 2.4. PBH Production in $f(R)$-Gravity

The idea of the proposed mechanism is based on the known possibility of formation of domain walls during cosmological inflation followed by their collapse into primordial black holes [13,20]. The formation of such domain walls requires a scalar field with a nontrivial potential containing several vacuums. It is this effective scalar field that arises in multidimensional $f(R)$-models in the Einstein frame [25,55,56]. This field controls the size of compact additional space, and its different vacuums correspond to different universes.

The model considered contains quadratic and tensor corrections to the scalar curvature:

$$S[g_{\mu\nu}] = \frac{m_D^{D-2}}{2} \int d^{4+n}x \sqrt{|g_D|} \left[ f(R) + c_1 R_{AB} R^{AB} + c_2 R_{ABCD} R^{ABCD} \right],$$

$$f(R) = a_2 R^2 + R - 2\Lambda_D, \tag{7}$$

The multidimensional space is represented as a direct product $\mathbb{M} = \mathbb{M}_4 \times \mathbb{M}_n$, where $\mathbb{M}_4$ is a four-dimensional space, $\mathbb{M}_n$ is a compact extra space, which is chosen in the form of a multidimensional sphere with $n$ dimensions:

$$ds^2 = g_{\mu\nu} dx^\mu dx^\nu - e^{2\beta(t)} d\Omega_n^2. \tag{8}$$

One can obtain the scalar curvature:

$$R = R_4 + R_n + P_k, \quad P_k = 2n\, \partial^2\beta + n(n+1)(\partial\beta)^2,$$

where $R, R_4, R_n$ are the scalar curvatures for $\mathbb{M}, \mathbb{M}_4$, and $\mathbb{M}_n$, respectively. As shown in [25], at the limit of the effective field theory:

$$R_4, P_k \ll R_n, \tag{9}$$

one obtain an effective field theory where the scalar curvature of extra space is considered a scalar field.

$$S = \frac{m_4^2}{2} \int d^4x \sqrt{-g_4}\, \text{sign}(f') \left[ R_4 + K(\phi)(\partial\phi)^2 - 2V(\phi) \right], \tag{10}$$

where the effective four-dimensional Planck mass in the Einstein frame is $m_4 = \sqrt{2\pi^{\frac{n+1}{2}}/\Gamma(\frac{n+1}{2})}$, $g_4^{\mu\nu}$ is the observable four-dimensional metric, and $f' = df/d\phi$.

The effective potential $V(\phi)$ happens to be appropriate for the domain wall formation. The kinetic term and the potential are given by the expressions [56]:

$$K(\phi) = \frac{1}{4\phi^2} \left[ 6\phi^2 \left( \frac{f''}{f'} \right)^2 - 2n\phi \left( \frac{f''}{f'} \right) + \frac{n(n+2)}{2} \right] + \frac{c_1 + c_2}{f'\phi}, \tag{11}$$

$$V(\phi) = -\frac{\text{sign}(f')}{2(f')^2} \left[ \frac{|\phi|}{n(n-1)} \right]^{n/2} \left[ f(\phi) + \frac{c_1 + 2c_2/(n-1)}{n} \phi^2 \right]. \tag{12}$$

The right minimum of potential (12) (Figure 2) corresponds to the observable universe [55,56], while the left minimum corresponds to the case of macroscopic extra dimensions, but due to the nontrivial kinetic term (11), it is impossible to reach the left minimum of the potential in finite time.

One can simplify the Lagrangian (10) by substituting

$$\psi = m_4 \int_{\phi_0}^{\phi} \sqrt{K(\phi')}\, d\phi', \quad \tilde{V}(\psi) = m_4^2 V(\phi(\psi)), \quad K(\phi) > 0, \tag{13}$$

and obtain a differential equation for the field evolution in a simple form. As mentioned above, due to nontrivial kinetic term (11), field $\phi$ and consequently $\psi$ are not able to reach the left minimum, so the lower limit in (13) is set to be $\phi_0 \ll \phi_{min}$, where $\phi_{min}$ corresponds to the right minimum of the potential. It is assumed that cosmological inflation is an external process, so there are inflationary constraints on the presented mechanism. As a result of repeated quantum fluctuations during cosmological inflation, the field $\psi(\phi)$ can be flipped from the rolling down to the right minimum to the region of rolling down to the left minimum in some area of the inflationary universe [20]. During inflation, the $\psi$

field is frozen near the potential maximum. After the end of inflation, the field tends to one minimum inside the bubble and another minimum outside of it. Increasing the energy density gradually forms the domain wall around the bubble.

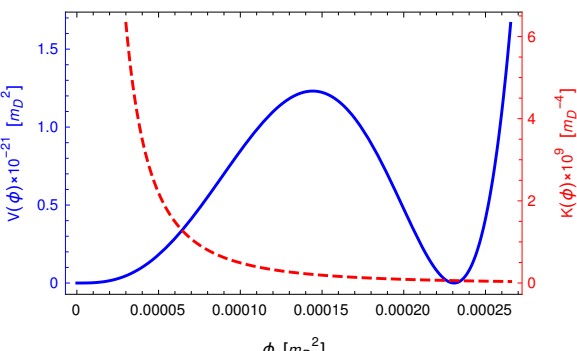

**Figure 2.** Graphs of potential (12) and kinetic terms (11) with the parameters chosen in [24]: $n = 6$, $c_1 = -8000, c_2 = -5000, a_2 = -500$.

It must be stated that $f(R)$-gravity adds a new possibility for the formation of black holes without any involvement of matter fields. It is also worth to mention that Planck collaboration data are well described by models of $f(R)$-gravity, e.g., [56,57], although the mass spectrum in this particular theory and possible observable properties are yet to be studied.

It also must be stated that since this model is based on the generation of domain walls via quantum fluctuations during the inflationary era, it has the same drawbacks and advantages as stated in the previous section. In that case, domain walls form due to quantum fluctuations during inflation, we also have a broad mass spectrum, see e.g., [13,50], and the mass spectrum in that case also strongly depends on the initial field value, so fine-tuning is required.

Further research should shed light on the mass spectrum in this theory. For example, it seems that the theory of modified gravity presented in this section could lead to very large and dense domain walls, so that they might be able to form baby universes.

## 3. Mass Accretion Mechanisms

### 3.1. Bondi Accretion

One of the first solutions to the accretion problem was that of Hoyle and Littleton [58], but the analytical formula was derived by Bondi [33]. Below is an overview of Bondi's solution.

The Bondi accretion is a spherically symmetric accretion on a compact object (Figure 3). The accretion rate is assumed to be $\dot{M} \approx \pi R^2 \rho v$, where $R$ is the capture radius or impact parameter, $\rho$ is the density of the surrounding matter, and $v$ is the the relative speed. The capture radius can be determined from the equality of the escape velocity and some characteristic velocity of matter. It is usually assumed to be equal to the speed of sound in the surrounding matter, then the accretion rate is obtained by $\dot{M} \approx \pi \rho G^2 M^2 / c_s^3$, where $c_s$ is the speed of sound in the matter surrounding the compact object.

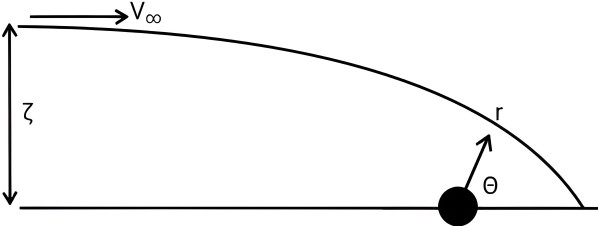

**Figure 3.** Schematic representation of the problem statement. $\zeta$ is the impact parameter and $v_\infty$ is the velocity of the test particle at a large distance from the accreting object.

The obvious drawback of the model is that it does not take into account possible relativistic effects, it (the model) is Newtonian. However, it is possible in this approach to take into account the expansion of the universe [59]. The Bondi problem can be posed as follows:

$$\dot{M} = 4\pi r^2 \rho v, \tag{14}$$

$$v\frac{dv}{dr} = -\frac{1}{\rho}\frac{dp}{dr} - \frac{GM(<r)}{r} - \beta(z)v, \tag{15}$$

$$p = K\rho^\gamma, \tag{16}$$

where $\beta(z)$ is the viscosity coefficient of the plasma around the accretor due to the interaction of electrons with photons, basically the Compton effect. The viscosity is given by the expression $\beta(z) = 2.06 \cdot 10^{-23} x_e (1+z)^4 c^{-1}$, where $x_e$ is the electron fraction in the plasma, and $z$ is the redshift. Term $M(<r)$ represents the mass contained within the radius $r$. The equation of state parameters are $K$ and $\gamma$, where $\gamma$ is a dimensionless constant. If we go to coordinates $r = a(t)x$, the Hubble expansion is added to the viscosity, and the effective viscosity is $\beta_{eff} = \beta(z) + H$. Thus, the rapid expansion of the universe reduces the accretion rate, and hence the accretor cannot significantly increase its mass.

It is necessary to note that Bondi's solution is designed to describe a stationary nonrelativistic accretion. There are also approaches that take into account the luminosity of the accreting object if one radiates [60] and that consider supermassive black hole accretion within the Bondi solution [61].

### 3.2. Accretion Inside Neutron Star

In the paper [62], the accretion by a small black hole trapped inside a neutron star is considered. With this mechanism, it is possible to give constraints on the population of small-mass black holes, i.e., to use neutron stars as "dark matter detectors". The observed population of neutron stars imposes constraints on the mass range of PBHs: $10^{-15} M_\odot \lesssim M_{PBH} \lesssim 10^{-9} M_\odot$.

As stated above, it is assumed that a black hole is trapped inside a neutron star and begins to absorb it. The equation of state of the matter inside the star is chosen in the form $P = K\rho^\Gamma$, which gives the speed of sound near the centre of the neutron star $a_c \approx \left(\frac{\Gamma P_c}{\rho_c}\right)^{1/2}$.

Then, the trapping radius of the black hole can generally be defined as:

$$r_c = G\frac{m(r_c) + M_{BH}}{a_c^2}. \tag{17}$$

If we set $m(r_c) \approx \frac{4\pi}{3} r_c^3 \rho_c$, then we can write the accretion rate as:

$$\dot{M} = 4\pi r_c^2 \rho_c a_c = 3a_c^3 G^2 \left(1 + \frac{M_{BH}}{m(r_c)}\right)^{-1}. \tag{18}$$

Using (17) and (18), we can write the characteristic accretion time:

$$\tau_{acc} = \frac{M_{BH}}{\dot{M}} = \frac{M_{BH}}{3a_c^3 G^2}\left(1 + \frac{M_{BH}}{m(r_c)}\right). \tag{19}$$

Note that in the limit $m(r_c) \ll M_{BH}$, it is reduced to the Bondi accretion.

This approach is mostly dedicated to impose a constraint on a specific mass range of PBHs, although it has a drawback: it is model-dependent since in order to calculate the speed of sound inside the star, one has to make an assumption about the equation of state

of the plasma inside the star. That means in order to impose constraints on PBHs, one has to make an assumption about the equation of state and its parameters.

Neutron stars should be studied in order to figure out the equation of state inside them.

### 3.3. Eddington Limit

The Eddington luminosity or Eddington limit is the maximum luminosity that an object can achieve given with an equilibrium (see Figure 4) between the gravitational force and the radiative pressure force. This state is called a hydrostatic equilibrium.

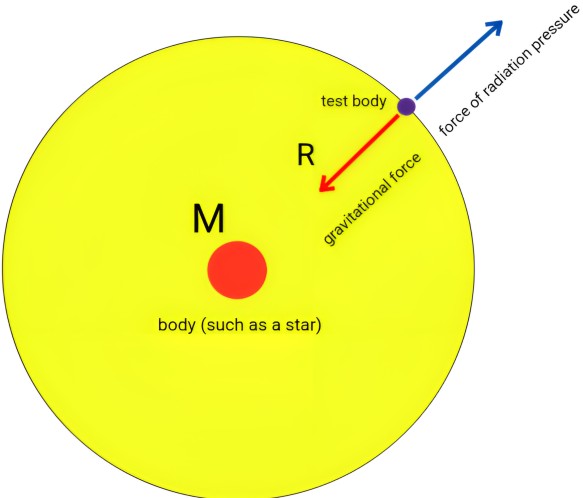

**Figure 4.** Eddington limit schematic representation.

The gravitational force acting on the test particle with mass $m$ is $F_{GRAV} = \dfrac{GMm}{R^2}$, and the radiation pressure is given by the formula $P_{RAD} = \dfrac{\Phi}{c} = \dfrac{1}{c}\dfrac{L}{f(R)}$, where $\Phi$ is the flux and the denominator is

$$f(R) = \int_{\phi=0}^{2\pi} \int_{\theta=-\pi/2}^{\pi/2} \int_{r=0}^{R} g(r,\theta,\phi)\, r^2 sin\theta dr d\theta d\phi, \tag{20}$$

where the integrand function characterises the distribution of matter within the object. Denote the plasma transparency by $\kappa$, then the radiation pressure force on the test object is $F_{RAD} = P_{RAD}\kappa m$. Equating the radiation pressure force to the gravitational force and assuming a spherical symmetry, that the main component of the plasma are protons, and the dominant process is the Thomson electron scattering, we obtain

$$L_{edd} = \frac{4\pi cGMm_p}{\sigma_{th}}. \tag{21}$$

The luminosity created by the accretion of matter can be represented in the following form $L_{acc} = \varepsilon \dot{M}c^2$, where $\varepsilon$ is the radiative efficiency. Equating the Eddington limit to the luminosity from the accretion of matter, we obtain:

$$L_{acc} = \varepsilon \dot{M}c^2 = L_{edd} = \frac{4\pi cGMm_p}{\sigma_{th}}. \tag{22}$$

It is easy to get the dependence of mass on time:

$$M(t) = M_0 \exp\left(\frac{4\pi Gm_p t}{\varepsilon c\sigma_{th}}\right). \tag{23}$$

Formula (23) is usually used as a marginal estimate of the mass change due to accretion, but in reality, this limit can be circumvented by the lack of spherical symmetry [63,64]; however, in the process of observing early quasars at redshifts $z\sim6$ [65], it was found that they were accreting at nearly the Eddington limit. It is also worth noting that the radiative efficiency $\varepsilon$ is generally referring to a function of the angular momentum of the accreting object. This dependence can significantly affect the black hole mass growth; see, e.g., [66], where the influence of the angular momentum on the accretion rate was considered in some detail.

Although originally it was derived for stars to limit their maximum mass, it is also applicable as an estimation of the accretion rate for accretion disks [67,68]. It also must be stated that realistic accretion disks models are too complicated and are utilized only via numerical simulations.

*3.4. Accretion in Schwarzschild Spacetime*

As an approach related to the general theory of relativity equations, the Schwarzschild metric is given by the expression:

$$ds^2 = -\left(1 - \frac{2GM}{r}\right)dt^2 + \frac{dr^2}{\left(1 - \dfrac{2GM}{r}\right)} + r^2 d\Omega^2, \tag{24}$$

where $d\Omega^2 = d\theta^2 + \sin^2\theta d\phi^2$. In [34,69], the (24) metric and the associated equality to zero of the covariant derivative of the energy–momentum tensor (EMT) were considered. The EMT was chosen as an ideal fluid as follows:

$$T_{\mu\nu} = (\rho + p)u_\mu u_\nu - p g_{\mu\nu}. \tag{25}$$

Omitting the details of the derivation given in [34,69], let us write down the expression for the accretion rate:

$$\dot{M} = 4\pi A G^2 M^2 (\rho_\infty + p_\infty), \tag{26}$$

where $A = \text{const}$, which, generally speaking, depends on the parameter of the equation of state $\omega$ ($p = \omega\rho$), and $\rho_\infty$ and $p_\infty$ are the energy density and pressure at infinity, respectively. The Equation (26) does not change its form for arbitrary $\omega$ (only the constant $A$ changes). For example, in [70], the RD (radiation-dominated epoch of the universe evolution with the equation of state parameter $\omega = 1/3$) stage was considered and the authors obtained the same equation, but with a different constant $A$.

Equation (26) is easily integrated if we put $\rho(t) = \rho_0(t_0/t)^2$, and we obtain:

$$1/M = 1/M_0 - 4\pi G^2 A \rho_0 t_0 (1 + \omega)\left(1 - \frac{t_0}{t}\right). \tag{27}$$

The obvious drawback of this model is that the Schwarzschild metric is not asymptotically Friedmann–Robertson–Walker (FRW), which casts doubt on the effectiveness of the resulting formula in the early stages of the evolution of the universe, when the characteristic accretion time is comparable to or greater than the Hubble time, i.e., the characteristic expansion time of the universe. There is an approach to match the cosmological Friedmann–Robertson–Walker metric with the local Schwarzschild metric in the context of evaluating the radiation accretion in an expanding universe [70].

*3.5. McVittie Solution*

In 1933, McVittie studied the influence of cosmic expansion on local physics [36] and derived a specific metric given by (28). The mechanism of generation of PBH proposed in [24] within the modified gravity framework possibly allows one to produce them practically immediately after the cosmological inflation. The question about the growth of

mass of a PBH with the evolution of the universe naturally arises. Since there are no observational data on this distant period of evolution of the universe, it is necessary to estimate the influence on the final result of the free parameters of the problem, namely, the parameter $\omega$ ($p = \omega\rho$) and the moment of the end of reheating $t_{reh}$. Of course, the initial mass of a black hole is also a free parameter; however, taking into account the fact that the early universe is still very rapidly expanding, it is necessary to consider an accretion model of matter taking into account the expansion of the universe. In [35,71], exact solutions to the accretion problem were considered. The solution with the McVittie metric and the energy–momentum tensor of a nonideal fluid with a radial flow is of most interest for the evaluation of the accretion in that case. In that model, it is possible to calculate how many times the mass of the PBH will change, as the mass of the PBH grows with the universe, depending on the state parameter $\omega$.

In the paper [35], the following metric was considered:

$$ds^2 = -\frac{B^2}{A^2}dt^2 + a^2(t)A^4\left(d\bar{r}^2 + \bar{r}^2 d\Omega^2\right),\tag{28}$$

where $A = 1 + \dfrac{Gm(t)}{2\bar{r}}$, $B = 1 - \dfrac{Gm(t)}{2\bar{r}}$. This metric is asymptotically FRW, and when the universe "stops" expanding, it passes into the Schwarzschild metric in isotropic coordinates via the radial coordinate substitution $\hat{r} = a\bar{r}$. Obviously, it describes a strongly gravitating object. In this spacetime, the physically relevant [72–74] mass will be the quasi-local mass $m_H(t) = m(t)a(t)$, and $m(t)$ is only the coefficient of the metric.

The EMT of matter around a black hole is

$$T_{ab} = (p + \rho)u_a u_b + p g_{ab} + q_a u_b + q_b u_a,\tag{29}$$

in which

$$u^a = \left(\frac{A}{B}\sqrt{1 + a^2 A^4 u^2}, u, 0, 0\right), \quad q^c = (0, q, 0, 0),\tag{30}$$

where $q^c$ describes the radial energy flux and $u^a$ is the four-velocity of the surrounding fluid. Then, the field equations are as follows:

$$\dot{m}_H = -aB^2 \mathcal{A}\sqrt{1 + a^2 A^4 u^2}[(p + \rho)u + q],\tag{31}$$

$$-3\left(\frac{AC}{B}\right)^2 = -8\pi\left[(p + \rho)a^2 A^4 u^2 + \rho\right],\tag{32}$$

$$-\left(\frac{A}{B}\right)^2\left(2\dot{C} + 3C^2 + \frac{2\dot{m}C}{\bar{r}AB}\right) = 8\pi\left[(p + \rho)a^2 A^4 u^2 + p + 2a^2 A^4 qu\right],\tag{33}$$

$$-\left(\frac{A}{B}\right)^2\left(2\dot{C} + 3C^2 + \frac{2\dot{m}C}{\bar{r}AB}\right) = 8\pi p,\tag{34}$$

where $\mathcal{A} = \int\int d\theta d\varphi\sqrt{g_\Sigma} = 4\pi a^2 A^4 \bar{r}^2$ and $C = \dfrac{\dot{a}}{a} + \dfrac{\dot{m}}{A\bar{r}}$. From the last two equations of the system, we obtain:

$$q = -(p + \rho)\frac{u}{2}.\tag{35}$$

Thus, we obtain the accretion rate:

$$\dot{m}_H = -\frac{1}{2}aB^2\sqrt{1 + a^2 A^4 u^2}(p + \rho)\mathcal{A}u.\tag{36}$$

Consider Formula (36). It is useful to find the dependence of the accretion rate on cosmological parameters; to do so, following [71], let us take the $\bar{r} \to \infty$ limit for expression (36):

$$\dot{m}_H = -2\pi a^3 (p_\infty + \rho_\infty) \lim_{r \to \infty} (ur^2).\tag{37}$$

It is interesting to compare this formula with others, such as the Babichev–Eroshenko–Dokuchaev formula [34,69,70]:

$$\frac{dM}{dt} = 4\pi G^2 A M^2 (p_\infty + \rho_\infty).\tag{38}$$

Formula (38) can be derived using the stationary Schwarzschild metric or the nonstationary Schwarzschild metric, in which metric coefficient $M$ is now a function of the time $M(t)$ [75]:

$$ds^2 = -\left(1 - \frac{2GM(t)}{r}\right)dt^2 + \left(1 - \frac{2GM(t)}{r}\right)^{-1} dr^2 + r^2 d\Omega^2.\tag{39}$$

Since the metric (28) changes into (39) at the "stop" expansion of the universe (in other words, we insert $\dot{a} = 0$ into the field equations) and replaces the radial coordinate, one would expect that the accretion rate (36) would change into Formula (38), but in general, this is not the case. The reason is that in deriving this formula, the assumption $\lim_{r \to \infty}(ur^2) = -2AG^2 M^2$ was made.

We continue to consider the limit $r \to \infty$ for Equation (36). For a large $\bar{r}$, we obviously have:

$$p(r;t) = p_\infty(t) + p_1(t)/r + \mathcal{O}(1/r^2),\tag{40}$$
$$\rho(r;t) = \rho_\infty(t) + \rho_1(t)/r + \mathcal{O}(1/r^2),\tag{41}$$
$$u(r;t) = u_\infty(t)/r^2 + \mathcal{O}(1/r^3).\tag{42}$$

Then, by substituting these approximations into the field equations and by combining the conservation laws, one can eventually obtain the dependence of $m_H(t)$ on the scale factor. See [71] for details of the derivation.

For the quasi-local mass, we obtain a second-order differential equation:

$$\ddot{m}_H + 2\frac{\dot{a}}{a}\dot{m}_H - 4\pi G[(3\omega + 1)m_H - 3\omega m_0 a](p_\infty + \rho_\infty) = 0.\tag{43}$$

The last expression contains a constant of integration $m_0$, which has not yet been assigned a definite meaning. The origin of this term is as follows: the differential equation for the first coefficient of the main part of the Laurent series for the density (40) can be derived from conservation laws. The equation for $\rho_1$ is as follows:

$$\dot{\rho}_1 + 3\frac{\dot{a}}{a}(p_1 + \rho_1) + 3G\dot{m}(p_\infty + \rho_\infty) = 0,\tag{44}$$

from which we get

$$\rho_1(t) = 3G(m_0 - m)(p_\infty + \rho_\infty).\tag{45}$$

Thus, an additional assumption about the value of $m_0$ at some initial point in time is required. Hence, it follows that (44) requires not two but three initial conditions to find a partial solution.

Thus, the dependence of the quasi-local mass on the scale factor is as follows:

$$m_H(t) = C_1 a(t)^{1+3\omega} - C_2 a(t)^{-3(1+\omega)/2} + \frac{3(1+\omega)}{3\omega + 5} m_0 a(t),\tag{46}$$

in which $C_1$ and $C_2$ are determined from the initial mass $m_H(t_0)$ and the initial accretion rate $\dot{m}_H(t_0)$. An example of using (46) is given in Figure 5.

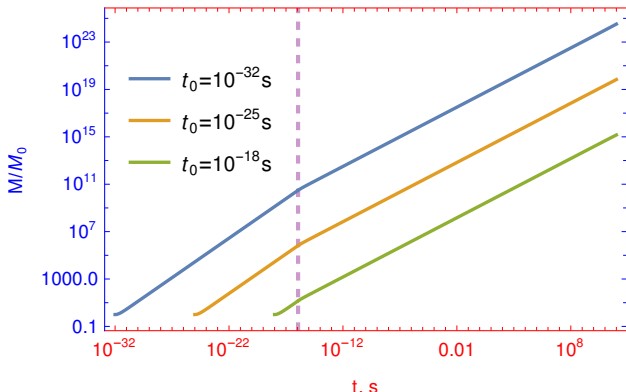

**Figure 5.** Relation between the initial mass of a PBH ($M_0$) at time $t$, which is $M$, derived from Formula (46). The time $t$ on the horizontal axis is the cosmic time. The dotted line shows the moment when the reheating ends ($10^{-16}$ s, chosen for demonstration); the equation of state parameter at the reheating stage was chosen as $\omega = 0$.

This model takes into account the cosmic expansion through the metric of the space-time, although this solution cannot be considered as realistic, since the fluid velocity on the sphere $\bar{r} = Gm(t)/2$ is superluminal (it is easy to see from Equation (36)).

## 4. Discussion

In the present review, we considered some mechanisms of the formation of PBHs such as the conventional mechanism based on the collapse of density fluctuations, phase transitions, e.g., first-order phase transitions which are strongly constrained by now. Moreover, the inflationary mechanism for domain wall production with their consequent collapse was considered, as well as the case of symmetry breaking with a consequent wall collapse. A recently elaborated mechanism based on the dynamic of extra space in a model of $f(R)$-gravity was also considered. They are mentioned in the following with a brief list of advantages and disadvantages:

- Initial density inhomogeneities:
  Advantages: a broad mass spectrum.
  Disadvantages: it leads to strong inhomogeneities and there is no possibility to form clusters of PBHs.
- First-order phase transitions:
  Advantages: it does not require any assumption beyond standard Big Bang physics.
  Disadvantages: The mass spectrum in this model is close to monochromatic, so it is impossible to explain the existence of black holes of various masses. It is also impossible to produce clusters of PBHs within this model.
- Second-order phase transitions:
  Advantages: it has a broad mass spectrum and it also could be coupled with external processes, e.g., baryogenesis.
  Disadvantages: a fine-tuning of the initial conditions is required.

The best-known black hole accretion models were also considered, such as the conventional Bondi solution (Newtonian approach), as well as a special case of the latter—accretion inside neutron stars. The Eddington limit as an approach of the marginal estimation of the accretion efficiency was also discussed. Next, a generalization of the Bondi accretion to the case of a curved Schwarzschild spacetime was considered. We also considered the accretion on a McVitty black hole, which was cosmologically coupled to the FRW universe. Here are the models and a brief list of their advantages and disadvantages:

- Bondi accretion:
  Advantages: a simple conventional model which is accurate enough.
  Disadvantages: it does not take into account possible relativistic effects and spacetime curvature.
- Accretion in Schwarzschild spacetime:
  Advantages: it takes into account spacetime curvature.
  Disadvantages: cosmic expansion is not considered in this case.
- McVittie solution:
  Advantages: it takes into account spacetime curvature with cosmic expansion.
  Disadvantages: it exhibits a superluminal motion of the fluid at a distance of $\bar{r} = Gm(t)/2$.

The topic of primordial black holes at the moment is a subject of great interest. They are potentially capable of explaining various cosmological problems, e.g., the origin of supermassive black holes, early structures' formation, and so on; thus, it is essential to elaborate and study the possibilities of their formation and evolution.

**Author Contributions:** Conceptualization, M.A.K. and V.V.N.; writing—original draft preparation, M.A.K.; writing—review and editing, V.V.N.; supervision, V.V.N. All authors have read and agreed to the published version of the manuscript.

**Funding:** The work of V.N. and M.K. was supported by the MEPhI Program Priority 2030.

**Institutional Review Board Statement:** Not applicable.

**Informed Consent Statement:** Not applicable.

**Data Availability Statement:** The original contributions presented in the study are included in the article, further inquiries can be directed to the corresponding author.

**Acknowledgments:** We greatly appreciate M. Yu. Khlopov for useful discussions.

**Conflicts of Interest:** The authors declare no conflict of interest.

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
