# Peer review of "Mechanisms of Producing Primordial Black Holes and Their Evolution"

_2571-712X, doi:10.3390/particles6020033_

Round 1

Reviewer 1 Report

This review paper recalls PBH production mechanisms and related mass accretion mechanisms. The paper is timely and collects a good number of relevant topics, which are well referenced. I do recommend the paper for publication in Particles. However, there are actions needed to improve the overall quality of the paper and even facilitate the understanding of a few steps in the explanations. I highlight the critical points using colours green and blue in the file attached. The parts highlighted in yellow generally deal with typos and minor missing definitions. It is highly desirable to fix at least the problems highlight in green and blue prior to publication. 

There is lack of "the" and other articles throughout the paper. Improvement on this aspect would be desirable, but not essential. I would recommend change the style of definition of quantities according to the following example: Instead of "z --- the redshift", it would be better "z is the redshift". There are multiple examples like this all over the manuscript.

Reviewer 2 Report

Dear authors, I read with great interest your manuscript "Mechanisms of producing PBHs and their evolution". Nevertheless, some questions appear and some commentaries must be improved. After that, I will decide it:

1.-On page 1:

-On the title, it is not correct to add an acronym (PBH) at the beginning. I suggest to change to Primordial Black Holes.

-On the abstract, your work is a paper or a review? Which is the novelty of the manuscript?

-The abstract is a little poor, please improve it.

2.-On page 2:

-Before section 2. Production mechanisms, I suggest adding a paragraph : "The plan of the work is organized as follows: In Section 2..."

-After section 2. Production mechanisms, I suggest an introductory paragraph.

-The paragraphs between lines 45 and 46 can be written in only one.

-On eq (2), what is the meaning of $R$?

-On line 49 there is a typo ("ans").

-On line 53, I suggest writing: "As a result, cosmological..."

-On line 68, I suggest writing: "was proposed by Khlopov et al. [41]."

-On eq (5), please explain the meaning of $\mu$.

3.-On page 4:

-On eq (6), what is the meaning of $\gamma$?.

-Between lines 124-125, I suggest writing: "One of the first papers dedicated to elaboration of this mechanism of PBH was performed in [46]".

-On line 132, please explain $\sigma$ and $G$.

-On eq (8), please explain $d\Omega_n^2$.

4.-On page 5:

-On eq (10), explain $f'$.

-On line 162 there is a typo.

-On eq (13), explain $\phi'$.

5.-On page 6:

-For the sake of completeness, I suggest adding a summary table, explaining these production mechanisms, with their advantages and disadvantages.

-On eq (15), what is the meaning of $GM(<r)$?.

6.-On page 7:

-On eq (17) a dot is required.

-On line 229, please explain better this part, is not clear.

-On line 233, I suggest changing "equilibrium (see Figure 4)."

7.-On page 8:

-On eqs (21) and (22) I suggest adding a dot.

8.-On page 9:

-On eq (24), please explain $d \Omega^2$.

-On line 260, I suggest changing to "The EMT is chosen as an ideal fluid as follows:"

-On line 266, please explain the RD stage.

9.-On page 10:

-On line 294, you say that the FRW passes into a Schwarzschild metric via radial coordinate replace, which is this transformation? 

-On line 298 there is a typo.

-Before eq. (31) I suggest changing to "Then, the field equations are as follows:"

-On eq (38), please change "[]" to "()".

10.-On page 11:

-Please on eq (39) explain that now $M(t)$ depends on $t$.

-On eq (46), please change "a^{1+3\omega}(t)" and "a^{-3(1+\omega)/2}(t)" to "a(t)^{1+3\omega}" and "a(t)^{-3(1+\omega)/2}".

11.-On page 12:

-At the end of Section 3, for the sake of completeness, I suggest adding a summary table, explaining these mass accretion mechanisms, with their advantages and disadvantages.

-On section 4, discussions:

(a)Again, your work is a review or a paper?

(b)What is the novelty of your work?

(c)With these review, which are the open problems or future works that can be raised from this information?

In my opinion, there are some typos in the document, but there are no issues with respect to the quality of the English Language broadly.

Author Response

The review is revised accordingly to your comments. Please see the attachment. 

Reviewer 3 Report

In this work the authors present a review of some

mechanisms for generating Primordial Black holes.

They focus on the description of the evolution of black holes via mass accretion. I consider the work is a comprehensive review and may be published in its present form

Author Response

I am grateful for your comments

Reviewer 4 Report

The work is interesting because in this form of paper, I recommended it for publication. I have thoroughly gone through the above titled manuscript and writing the following report: The problem is very interesting. This paper has nice motivation and lucid presentation with reasonably good English. This work, I am sure, will go a long way to inspire researchers in this particular field. Hence, I recommend this article for publication in the present form in standard Particles journal.

Author Response

I am grateful for your comments

Round 2

Reviewer 2 Report

Dear authors, thanks for considering the commentaries and suggestions from the first round. In my opinion, your work is adequate to be published in particles.